# Weight Loss and Short-Chain Fatty Acids Reduce Systemic Inflammation in Monocytes and Adipose Tissue Macrophages from Obese Subjects

**DOI:** 10.3390/nu14040765

**Published:** 2022-02-11

**Authors:** Shaun Eslick, Evan J. Williams, Bronwyn S. Berthon, Timothy Wright, Constantine Karihaloo, Mark Gately, Lisa G. Wood

**Affiliations:** 1Priority Research Centre for Healthy Lungs, Hunter Medical Research Institute, University of Newcastle, New Lambton Heights, Newcastle, NSW 2305, Australia; shaun.eslick@uon.edu.au (S.E.); evan.j.williams@newcastle.edu.au (E.J.W.); bronwyn.berthon@newcastle.edu.au (B.S.B.); 2Department of Surgery, John Hunter Hospital, New Lambton Heights, Newcastle, NSW 2305, Australia; admin@drtimwright.com.au (T.W.); costa_k@mac.com (C.K.); 3Sea Change Weight Loss Clinic, Merewether, Newcastle, NSW 2291, Australia; mcgately@gmail.com

**Keywords:** obesity, systemic inflammation, weight loss, short-chain fatty acids, FFAR, HDAC

## Abstract

Background: Chronic low-grade systemic inflammation is a characteristic of obesity that leads to various non-communicable diseases. Weight loss and SCFAs are potential strategies for attenuating obese systemic inflammation. Methods: Blood samples were collected from 43 obese subjects (BMI ≥ 30 kg/m^2^) scheduled for laparoscopic bariatric sleeve surgery, 26 obese subjects at follow-up 12–18 months post-surgery and 8 healthy weight subjects (BMI 18.5–24.9 kg/m^2^). Monocytes were isolated from blood and adipose tissue macrophages from visceral adipose tissue of obese subjects only. Isolated cells stimulated with 1 ng/mL LPS and treated simultaneously with 300 mM of sodium acetate or 30 mM of sodium propionate or butyrate and supernatant were harvested after 15 h incubation. TNF-α and IL-6 cytokines were measured via ELISA and mRNA gene expression of *FFAR2* and *FFAR3*, *HDAC1*, *HDAC2* and *HDAC9*, *RELA* and *NFKB1* and *MAPK1* via RT-qPCR. Results: TNF-α and IL-6 production and *NFKB1* and *RELA* mRNA expression were significantly decreased in follow-up subjects compared to baseline. SCFAs significantly reduced TNF-α and IL-6 and altered *FFAR* and *HDAC* mRNA expression in monocytes and macrophages from obese subjects. Conclusion: Weight loss and ex vivo SCFA treatments were successful in combatting systemic inflammation in obesity. Results highlighted molecular changes that occur with weight loss and as a result of SCFA treatment.

## 1. Introduction

Chronic low-grade systemic inflammation is a hallmark feature of obesity. Systemic inflammation in obesity can contribute to the development of a number of non-communicable chronic diseases such as type 2 diabetes (T2D), liver disease, arthritis and some types of cancer [1].

In obesity, systematic inflammation is largely driven by adipose tissue macrophages (ATMs). As a consequence of excess energy intake, significant enlargement of adipose tissue occurs, stimulating polarisation of macrophages from an anti-inflammatory M2 phenotype to a pro-inflammatory M1 phenotype [2]. M1 macrophages exacerbate systemic inflammation by production of pro-inflammatory cytokines such as TNF-α and IL-6 [3]. Moreover, M1 macrophages become unable to buffer lipids, resulting in a lipid spillover into circulation, activating pro-inflammatory cascades [3]. Secondly, obese systemic inflammation can be driven by metabolic endotoxemia, a state typified by elevated levels of circulating endotoxins including lipopolysaccharides (LPS). In obesity, metabolic endotoxemia occurs as a result of gut dysbiosis, consequently altering tight junction proteins in the intestinal lumen, increasing intestinal permeability, permitting LPS to infiltrate circulation [4]. Elevated levels of metabolic and bacterial danger signals such as free fatty acids (FFAs) and LPS initiate systemic inflammation by binding to Toll-like receptors (TLRs) and activating NF-κB and MAPK pro-inflammatory pathways leading to transcription of pro-inflammatory cytokines such as TNF-α, IL-6 and IL-1β [5,6].

One of the most effective strategies for combating chronic low-grade systemic inflammation in obesity is weight loss [7]. Bariatric surgery induces significant weight loss, resulting in significant reductions in systemic inflammation [8], with a previous study showing a significant reduction in pro-inflammatory cytokines CRP, TNF-α and IL-6 [9]. This highlights weight loss as an important strategy for attenuating systemic inflammation in obesity, possibly contributing to the reduction in chronic diseases such as T2D seen in this population.

A paucity of research has examined changes to pro-inflammatory pathways in obesity as well as following significant weight loss. It is hypothesised that NF-κB and MAPK pro-inflammatory pathways are upregulated in obesity, promoting production of pro-inflammatory cytokines that contribute to the development of chronic disease [5,10]. As significant reductions in pro-inflammatory cytokines are seen with significant weight loss following bariatric surgery, it is hypothesised that downregulation of NF-κB and MAPK pathways also occurs. 

An alternate, novel and non-invasive treatment for systemic inflammation in obesity is short-chain fatty acids (SCFAs). SCFAs are metabolically active end products formed by fermentation of prebiotic soluble fibre by intestinal microbes [11]. The majority of SCFA production occurs within the caecum and proximal segments of the colon [12], with acetate, propionate and butyrate making up >95% of SCFA content [11]. In circulation, acetate is primarily found in the highest concentrations (19–160 µmol/L), whilst butyrate and propionate are found in lesser concentrations (1–13 and 1–12 µmol/L, respectively) [13].

Evidence suggests that SCFAs can inhibit LPS-induced inflammation, a primary trigger of systemic inflammation in obesity [14]. A number of mechanisms have been suggested to explain the anti-inflammatory potential of SCFA [6,15]. Firstly, by their action on free fatty acid receptor (FFAR) 2, which is primarily expressed on immune cells and FFAR3 which is predominantly expressed in the pancreas, spleen and adipose tissue and is therefore associated with obesity and metabolic diseases such as T2DM [6]. FFAR2 and 3 are upregulated by LPS stimulation in monocytes and macrophages further highlighting the potential role of FFARs in systemic inflammation in obesity [16]. Secondly, SCFAs can inhibit histone deacetylases (HDACs), resulting in downregulation of NF-κB and MAPK pro-inflammatory pathways [6]. SCFAs butyrate and propionate interact predominately with HDACs 1 and 2 located inside the nucleus, in addition to HDACs 3, 4, 5, 6, 7 and 9 that shuttle between the cytosol and the nucleus [17]. Butyrate and propionate, proposed to be the most potent HDAC inhibitors, have previously been shown to suppress TNF-α production and NF-κB activity, whilst evidence indicates that acetate has little ability to inhibit HDAC activity [18,19,20].

This study aimed to investigate two strategies for reducing chronic low-grade systemic inflammation in obesity. Firstly, this study aimed to characterise changes in systemic inflammation that occur with weight loss following bariatric surgery. Secondly, this study utilised an ex vivo model to assess the potential of SCFAs as a novel anti-inflammatory treatment for systemic inflammation in obesity. This model assessed pro-inflammatory cytokine response and changes to molecular pathways in peripheral blood monocytes and ATMs following SCFA treatment.

## 2. Materials and Methods

### 2.1. Study Design

Obese adults (*n* = 43) (BMI ≥ 30 kg/m^2^) were recruited from obesity surgery clinics, confirmed for laparoscopic sleeve gastrectomy at Lake Macquarie Private Hospital, Newcastle Private Hospital and Lingard Hospital, NSW, Australia between May 2018 and October 2019. Collaborating surgeons introduced this study to prospective subjects during pre-surgery consultation, whereby prospective subjects gave consent to be contacted about inclusion in this study. Consenting subjects were contacted for a telephone screening interview to determine eligibility. Eligibility for this study required being an adult (18 years or older) with a BMI ≥ 30 kg/m^2^, not a current smoker and being scheduled for a sleeve gastrectomy. Exclusion criteria included; having signs of active inflammation such as an autoimmune disease, using systemic anti-inflammatory medications (oral corticosteroids, disease modifying anti-rheumatic drugs) and current smoking (within 12 months). This study was performed at the Hunter Medical Research Institute (HMRI), New Lambton Heights, NSW, Australia. All subjects gave written informed consent and this study was approved by the Hunter New England Human Research Ethics Committee (New Lambton, Australia; reference no: 13/07/17/4.03). Subjects were assigned a very low calorie diet (VLCD) by their operating surgeon, to consume 1–4 weeks prior to surgery. VLCD regimes varied amongst collaborating surgeons and individual subjects. Obese adults (*n* = 26) recruited prior to bariatric surgery attended a follow up clinic visit 12–18 months post-bariatric surgery. Adult volunteers (*n* = 8) from HMRI were recruited to form a healthy weight control cohort (BMI < 30 kg/m^2^), and attended a single clinical visit.

### 2.2. Anthropometric Measurements

Body weight (kg) and height were measured in light clothing. BMI was calculated using body weight (kg)/height (m^2^). Body composition was measured using dual energy X-ray absorptiometry (DEXA) machine and its coupled software (DEXA Lunar Prodigy; Encore 2007 Version 11.40.004, GE Medical Systems, Madison, WI, USA). Total and regional fat mass and lean mass were reported in grams and fat percentage was calculated as (total fat (g)/total body weight (g)) × 100. 

### 2.3. PBMCs and Monocyte Isolation from Whole Blood

Whole blood was collected in 2 × 9 mL ethylenediamine-tetraacetic acid (EDTA) tubes. PBMCs were isolated from whole blood using SepMate tubes (STEMCELL Technologies, Vancouver, BC, Canada) as per manufacturer’s specifications. Monocytes were isolated from PBMCs using MACS™ Monocyte Isolation Kit II (Miltenyi Biotech, Cologne, Germany) as per manufacturer’s specifications. Isolated monocytes were counted on a haemocytometer and assessed for cell viability using 0.4% trypan blue for staining, then suspended in DMEM (Sigma Aldrich, St. Louis, MO, USA) with 10% foetal bovine serum (Bovogen Biologicals, East Keilor, Victoria, Australia) and plated in 96 well plates (Thermo Scientific Nunc Scoresby, Victoria, Australia) at a final concentration of 5.0 × 10^4^ live cells per well in 250 μL of media.

### 2.4. Macrophage Isolation from Adipose Tissue

Adipose tissue isolation and analysis were performed in obese subjects only, where excision was carried out by qualified bariatric surgeons during laparoscopic gastric sleeve surgery. Adipose tissue was collected from omental fat depots and biopsies were taken from regions considered medically safe by the operating surgeon and were consistent across each subject. Visceral adipose tissue was cut from free margins of omental fat, far from blood vessels. The sample was resected using a harmonic scalpel and transferred immediately into a sterile specimen jar that contained 20 mL DMEM (Dulbecco’s modified Eagle Medium, Sigma-Aldrich, St. Louis, MO, USA). Adipose tissue was transported on ice to HMRI lab for processing within an hour of excision.

To dissect visceral adipose tissue, visible blood vessels and connective tissue were removed using sterile forceps and scissors. Up to 8 g of visceral adipose tissue (4 g maximum per tube) was weighed and then washed with 20 mL DPBS and minced with scissors into pieces approximately 1–2 mm in size. Minced samples were centrifuged at 546× *g* with DPBS at 4 °C for 10 min to remove red blood cells. A digestion mixture consisting of 20 mL DMEM supplemented with 0.2 g of fatty acid poor BSA (bovine serum albumin lyophilised powder, Calbiochem, San Diego, CA, USA), 1 µg/mL Collagenase Type I (Worthington Biochemical, Lakewood, NJ, USA) and 60 units/mL of DNAse I (Sigma-aldrich, St. Louis, MO, USA) was added for each 4 g of adipose tissue and incubated at 37 °C for 1 h. The sample was then filtrated through a strainer, into a new tube and centrifuged at 612× *g* at 4 °C for 10 min. Supernatant was discarded and the remaining pellet was re-suspended in 5 mL erythrocyte lysis buffer (Qiagen, Hilden, Germany) for 5 min at room temperature. The sample was then passed through a 70 µm cell strainer and once again centrifuged at 612× *g* at 4 °C for 10 min. Supernatant was removed and macrophages were isolated from SVF using CD14^+^ microbeads (MACS, Miltenyi Biotech, Cologne, Germany) as per manufacturer’s instructions. Isolated macrophages were counted and assessed for cell viability using 0.4% Trypan Blue, then resuspended in Dulbecco’s Modified Eagles Medium High Glucose (Sigma Aldrich, St. Louis, MO, USA) with 10% foetal bovine serum (Bovogen Biologicals, East Keilor, Victoria, Australia) and plated in 96 well plates (Thermo Scientific Nunc Scoresby, Victoria, Australia) at a final concentration of 1.0 × 10^5^ cells per well. 

### 2.5. In Vitro Treatment and Culture of Peripheral Blood Monocytes and Adipose Tissue Macrophages with Short-Chain Fatty Acids

Plated cells were simultaneously treated with LPS (Monocytes: 1 ng/mL, macrophages: 100 ng/mL) and 30 mM of sodium propionate or butyrate solubilised in Dulbecco’s Phosphate-Buffered Saline (DPBS, Sigma Aldrich, St. Louis, MO, USA) or 300 mM of sodium acetate solubilised in DMEM with 10% foetal bovine serum (Bovogen Biologicals, East Keilor, Victoria, Australia) and incubated for 15 h. Sodium acetate was solubilised in DMEM with 10% foetal bovine serum due to treatments requiring higher concentrations of stock solution. Concentrations of SCFA treatments were determined from published literature [21,22,23,24]. Additionally, DMEM media-only control and LPS-stimulated (LPS and media) controls were also cultured for 15 h. Cell culture supernatant was collected after 15 h from all experiments.

### 2.6. Cell Culture and Serum TNF-α and IL-6 ELISA

Cytokine measurement in human serum was performed using Human TNF-α and IL-6 Quantikine ELISA kits (R&D Systems, Minneapolis, MN, USA) as per manufacturer’s specifications. Cytokines in cell supernatant were measured using Human TNF-α and IL-6 DuoSet^®^ ELISA kits (R&D Systems, Minneapolis, MN, USA) as per manufacturer’s specifications.

### 2.7. mRNA Expression

RNA was collected by lysing cells in RLT buffer with 0.1% 2-mercaptoethanol (Qiagen, Hilden, Germany). RNA extraction was performed using RNeasy mini kit (Qiagen, Hilden, Germany) as per manufacturers specifications.

NanoDrop 2000 spectrophotometer (Thermo Scientific, Waltham, MA, USA) was used for quantification of RNA. Real time quantitative PCR using standard taqman probes and methods (Applied Biosystems, Foster City, CA, USA) was undertaken to assess mRNA expression of HDAC (Histone Deacetylase) 1, 2, and 9, NFκB1 (nuclear factor kappa-light-chain-enhancer of activated B cells), RELA (v-rel avian reticuloendotheliosis viral oncogene homolog A), FFAR (free fatty acid receptor) 2, and 3, and MAPK (Mitogen-activated protein kinases) 1. Extracted RNA (30 ng) was converted into cDNA using High Capacity cDNA Reverse Transcription Kit (Applied Biosystems, Foster City, CA, USA). Gene expression targets were measured relative to housekeeping gene 18 s. All reactions were performed using QuantStudio 6 Flex Real-Time PCR System (Thermo Fisher, Waltham, MA, USA). Taqman assay ID’s; HDAC1 (Hs02621185_s1), HDAC2 (Hs00231032_m1), HDAC9 (Hs01081558_m1), NFκB1 (Hs00765730_m1), RELA (Hs01042014_m1), FFAR2 (Hs00271142_s1), FFAR3 (Hs02519193_g1), MAPK1 (Hs01046830_m1). mRNA expression values were presented as 2-ΔCT, where ΔCT = Target CT − 18 s CT. 

### 2.8. Statistical Analysis

Statistical analysis was completed using GraphPad Prism Version 9.2.0 (GraphPad Software, San Diego, CA, USA). Parametric data were reported as the mean ± standard deviation (SD), whilst non-parametric data were presented as the median (1st quartile, 3rd quartile). Normality of data was assessed via Shapiro–Wilk, where normality was indicated by *p*-value > 0.05. *p*-values < 0.05 were considered significantly significant. Repeated-measures (RM) one-way ANOVA or the Friedman test was used to compare TNF-α, IL-6 and qPCR targets in monocytes and macrophages where appropriate. Tukey’s multiple comparison test was used for post hoc testing when RM one-way ANOVA was performed and Dunn’s multiple comparisons test used to perform post hoc testing when the Friedman test was utilised. Sample sizes differed in monocytes and macrophages due to varying numbers of monocytes and macrophages that were able to be isolated from each sample.

## 3. Results

### 3.1. Subject Characteristics

Data from 43 obese subjects scheduled for bariatric surgery were compared with 8 healthy control subjects (Table 1). Subjects were predominantly female in both groups. The obese group had a median BMI of 46.4 kg/m^2^ and had a mean age of 41.6 years. As expected, BMI was significantly lower in healthy controls (med: 22.3 kg/m^2^, *p* < 0.001), who were also younger (med: 28.9 years, *p* < 0.001) than the obese cohort. Furthermore, the healthy cohort had significantly lower % total fat (*p* < 0.001) and higher lean mass (*p* < 0.001) compared to the obese cohort.

A total of 26 obese subjects attended a follow-up visit, on average 15 months after bariatric surgery. At follow up, subjects had significantly lower mean weight (*p* < 0.001), BMI (*p* < 0.001), % fat mass (*p* < 0.001), and significantly higher mean % lean mass (*p* < 0.001).

### 3.2. Presence of Systemic Inflammation in Obesity

Serum TNF-α (Figure 1a) was significantly higher in the obese group compared to the healthy control group (*p* = 0.002). Serum IL-6 (Figure 1b) in obese subjects trended towards a significant increase compared to healthy control subjects (*p* = 0.075).

*NFKB1* mRNA expression (Figure 1c) was significantly higher in stimulated monocytes from obese subjects compared to healthy stimulated monocytes (*p* < 0.001). *RELA* mRNA expression (Figure 1d) was significantly higher in stimulated monocytes from obese subjects, compared to those from healthy subjects (*p* = 0.001). No differences were observed in *MAPK1* mRNA expression (Figure 1e) between obese and healthy stimulated monocytes (*p* = 0.32).

### 3.3. Changes to Systemic Inflammation following Weight Loss

Serum TNF-α (Figure 2a) was significantly lower at follow up compared to serum taken prior to bariatric surgery (*p* < 0.001). Similarly, serum IL-6 (Figure 2b) significantly decreased at follow up compared to prior surgery (*p* = 0.001).

A significant decrease was observed in the expression of NF-κB-related genes *NFKB1* (*p* = 0.019) (Figure 2c) and *RELA* (*p* = 0.017) (Figure 2d) in stimulated monocytes at follow-up compared to stimulated monocytes before bariatric surgery. No change in *MAPK1* mRNA expression in stimulated monocytes before and after bariatric surgery (*p* = 0.383) (Figure 2e) was observed.

### 3.4. Pro-Inflammatory Cytokine and Gene Expression Response of LPS-Stimulated Peripheral Blood Monocytes to Short-Chain Fatty Acid Treatments

LPS-stimulated monocytes produced significantly higher TNF-α (*p* < 0.001) and IL-6 (*p* < 0.001) than media-only treated monocytes confirming that inflammation was induced by LPS-stimulation. 300 mM acetate treatment (*p* < 0.001), 30 mM propionate (*p* < 0.01) and 30 mM butyrate (*p* < 0.001) significantly reduced TNF-α (Figure 3a) and IL-6 (Figure 3b) compared to untreated, LPS-stimulated monocytes.

LPS-stimulated monocytes treated with acetate and propionate did not significantly change *FFAR2* mRNA gene expression compared to untreated, LPS-stimulated monocytes (Figure 4a). *FFAR2* mRNA expression was, however, significantly decreased in LPS-stimulated monocytes treated with butyrate compared to LPS-stimulated-only monocytes (*p* < 0.001). A non-significant increase in *FFAR3* mRNA expression was observed in LPS-stimulated monocytes treated with acetate (*p* = 0.124) compared to LPS-stimulated-only monocytes, whilst propionate and butyrate did not significantly change *FFAR3* mRNA expression (Figure 4b). 

SCFAs did not alter *HDAC1* mRNA expression compared to LPS-stimulated-only monocytes (Figure 4c). Acetate (*p* = 0.013) and butyrate (*p* < 0.001) both significantly reduced *HDAC2* mRNA expression compared to LPS-stimulated-only monocytes, whilst propionate had no effect (Figure 4d). LPS stimulation significantly increased *HDAC9* mRNA expression in monocytes (*p* < 0.05) (Figure 4e). Acetate treatment did not affect mRNA expression of *HDAC9*, whilst propionate (*p* < 0.001) and butyrate (*p* < 0.001) treatments significantly decreased LPS-induced *HDAC9* expression compared to LPS-stimulated-only monocytes. 

NF-κB associated gene, *NFKB1* mRNA expression was downregulated in LPS-stimulated monocytes treated with acetate (*p* = 0.013) and butyrate (*p* < 0.001) compared to LPS-stimulated-only monocytes (Figure 4f). LPS-stimulated monocytes treated with propionate did not change *NFKB1* mRNA gene expression. NF-κB associated gene *RELA* mRNA expression remained unchanged following acetate and propionate SCFA treatments (Figure 4g). *RELA* mRNA gene expression significantly decreased in LPS-stimulated monocytes treated with butyrate compared to LPS-stimulated-only monocytes (*p* < 0.001).

mRNA gene expression of *MAPK1* was not affected by acetate or propionate treatments compared to LPS-stimulated-only monocytes (Figure 4h). Butyrate treatment significantly decreased *MAPK1* compared to LPS-stimulated-only (*p* < 0.001) monocytes.

### 3.5. Pro-Inflammatory Cytokine and Gene Expression Response of LPS-Stimulated Adipose Tissue Macrophages to Short-Chain Fatty Acid Treatments

TNF-α production was significantly reduced in LPS-stimulated macrophages treated with 300 mM acetate (*p* = 0.039) and 30 mM butyrate (*p* = 0.003) compared to untreated, LPS-stimulated macrophages (Figure 5a). Treatment of LPS-stimulated macrophages with propionate did not alter TNF-α production. IL-6 was not significantly reduced by acetate treatment in macrophages (Figure 5b). In contrast, LPS-stimulated macrophages treated with 30 mM propionate (*p* < 0.001) and butyrate (*p* < 0.001) showed significantly lower IL-6 production compared to untreated, LPS-stimulated macrophages.

In LPS-stimulated macrophages, acetate treatment trended towards a significant increase in *FFAR2* mRNA expression compared to LPS-stimulated-only monocytes (*p* = 0.0733) (Figure 6a). No changes in *FFAR2* mRNA expression was seen as a result of propionate and butyrate treatments. Acetate treatment significantly upregulated *FFAR3* mRNA expression compared to LPS-stimulated-only macrophages (*p* = 0.010) (Figure 6b). Similar to FFAR2, *FFAR3* mRNA expression was not changed following propionate and butyrate SCFA treatments.

Acetate significantly upregulated *HDAC1* mRNA expression compared to LPS-stimulated-only (*p* = 0.040) macrophages (Figure 6c). *HDAC1* mRNA expression was not altered by propionate treatment. In contrast, *HDAC1* mRNA expression was significantly higher in LPS-stimulated macrophages treated with butyrate compared to LPS-stimulated-only (*p* = 0.018) macrophages. *HDAC2* mRNA expression was unchanged as a result of SCFA treatments (Figure 6d). *HDAC9* mRNA expression was not affected by acetate, whilst propionate (*p* = 0.017) and butyrate (*p* = 0.007) significantly downregulated *HDAC9* mRNA expression compared to LPS-stimulated-only macrophages (Figure 6e).

Acetate (*p* = 0.043), propionate (*p* = 0.002) and butyrate (*p* = 0.006) SCFA treatments all significantly downregulated NF-κB associated gene *NFKB1* in LPS-stimulated macrophages compared to LPS-stimulated-only macrophages (Figure 6f). NF-κB associated gene *RELA* mRNA expression in LPS-stimulated macrophages was significantly reduced by acetate treatment compared to LPS-stimulated-only macrophages (*p* = 0.018), whilst propionate and butyrate had no effect (Figure 6g). Finally, MAPK associated gene *MAPK1* mRNA gene expression in LPS-stimulated macrophages was not altered by SCFA treatment (Figure 6h).

## 4. Discussion

This study presents an analysis of systemic inflammation in obesity, the molecular pathways driving inflammation, and explores two strategies for attenuating inflammation in obesity, including bariatric surgery and SCFA treatment. This analysis highlighted an increase in pro-inflammatory cytokines TNF-α and IL-6 and upregulation of NF-κB-related genes NFKB1 and RELA in obesity. Secondly, we investigated changes in systemic inflammation following weight loss induced by bariatric surgery, which induced significant loss of body and fat mass. We observed a significant reduction in serum cytokines TNF-α and IL-6 as well as downregulation in mRNA expression of NF-κB-related genes NFKB1 and RELA in LPS-stimulated monocytes compared to baseline. Finally, we examined the effect of SCFA treatment, specifically, acetate, butyrate and propionate, in an ex vivo model of LPS-induced inflammation in immune cells derived from obese subjects. SCFA treatments in LPS stimulated monocytes and ATMs significantly reduced inflammatory cytokines TNF-α and IL-6. Furthermore, SCFA treatments were found to modulate the expression of FFAR and HDAC genes involved in key inflammatory pathways in both monocytes and ATMs from obese subjects.

NF-κB and MAPK pro-inflammatory pathways are regulators of systemic inflammation in obesity [25]. Activation of the NF-κB and MAPK pathways in obesity occurs due to exposure to immune cells to increased circulating saturated fatty acids and LPS, increasing production of pro-inflammatory cytokines, including TNF-α and IL-6 [26]. Recent evidence from a study in mice revealed that gut dysbiosis as a result of high energy diet leads to an increase in circulating inflammatory markers [27]. In this study, serum TNF-α levels were higher in obese than healthy weight subjects, in agreement with previous literature [28]. IL-6 levels were also higher in obese subjects, though the difference in IL-6 did not reach significance due to sample size. This finding is also supported by evidence from the literature showing higher serum IL-6 in obese subjects [28,29].

Importantly, differences in pro-inflammatory gene expression were identified in stimulated cells from obese and healthy subjects. NF-κB-related genes NFKB1 and RELA were upregulated in LPS-stimulated monocytes in obese versus healthy weight subjects. Though there was no difference in MAPK-related gene MAPK1 expression. Few studies have analysed NFKB1, RELA and MAPK1 mRNA expression in obesity. Sheu et al. found mRNA expression of RELA and p105 (subunit of NFKB1) were significantly higher in mononuclear cells isolated from obese women compared to lean women [6,30]. NFKB1 (p50) and RELA (p65), components of NF-κB, form a heterodimer, where upon activation of TLRs by FFAs or LPS, translocation of the NF-κB heterodimer to the nucleus occurs, where it binds to DNA, activating gene transcription of pro-inflammatory cytokines [5]. Hill et al. found that the NF-κB pathway plays a key role in adipose tissue recruitment of monocytes/macrophages in obesity and drives the recruitment and survival of pro-inflammatory M1 macrophages via NF-κB-dependent genes/proteins [31]. In contrast, MAPK1 mRNA expression was not affected by weight status in this analysis. These results suggest that the NF-κB pro-inflammatory pathway may play a more important role in systemic inflammation in obesity relative to the MAPK pathway.

Bariatric surgery is a successful intervention for inducing weight loss and subsequent fat mass loss in obesity. Indeed, comparison of body composition measurements 12–18 months follow-up to baseline measurements highlighted significant decreases in BMI, weight and total fat mass and a significant increase in lean mass within our cohort. This study showed that mean serum TNF-α and IL-6 were significantly reduced following weight loss induced by bariatric surgery. A recent meta-analysis by Askarpour et al. showed that 12 months following all types of bariatric surgery, TNF-α and IL-6 were significantly reduced [9]. Pooled results showed larger changes in IL-6 and TNF-α compared to this study; however, considerable heterogeneity was seen in both pooled analyses, most likely due to inclusion of differing bariatric surgery procedures, cohorts and time of follow up [9].

In addition to changes in pro-inflammatory cytokines, this study showed significant downregulation in gene expression of pro-inflammatory pathways. As aforementioned, obesity and increased adipose tissue volume coincide with elevated expression of pro-inflammatory pathways NF-κB and MAPK compared to lean subjects. Therefore, downregulation of mRNA gene expression of pro-inflammatory pathways following weight loss can be expected. Results revealed that gene expression of NF-κB-related genes NFBK1 and RELA by LPS stimulated cells was significantly reduced with weight loss at 12–18 months post-bariatric surgery. Similarly, Sheu et al. observed significant decreases in relative RELA and p105 mRNA gene expression in obese women following weight loss achieved by dietary intervention [30]. To date, the effects of weight loss on the MAPK pathway has not been investigated. mRNA gene expression of MAPK1 was not significantly upregulated in obese subjects compared to healthy controls, therefore it may be expected that a significant downregulation of MAPK1 would not observed following weight loss. These findings further reinforce the notion that the NF-κB inflammatory pathway may have a more dominant role in systemic inflammation in obesity compared to MAPK inflammatory pathway. Current literature proposes that the NF-κB and MAPK pro-inflammatory pathways play a pivotal role in systemic inflammation in obesity; however, few studies have investigated the role of NF-κB- and MAPK-related gene expression. Future studies should aim to investigate these pathways to further elucidate their role in development and maintenance of systemic inflammation in obesity.

This study showed that in LPS-stimulated monocytes isolated from obese subjects, butyrate, propionate and acetate all significantly reduced production of TNF-α and IL-6. Production of TNF-α from LPS-stimulated ATMs was significantly reduced by acetate and butyrate and IL-6 by propionate and butyrate. Tedelind et al. found that acetate, propionate and butyrate all decreased LPS-induced TNF-α response in human monocytes from healthy subjects [22]. Similarly, Fukae et al. demonstrated a significant decrease in LPS-stimulated TNF-α with treatments of 2 and 10 mM of sodium butyrate in monocytes from healthy humans [32]. While Cox et al. showed that SCFAs (0.2 mmol/L to 20 mmol/L) were effective in decreasing LPS-induced TNF-α in PBMCs, this was not the case in human monocytes [23]. Few studies have measured the effects of SCFAs on obese ATMs. Al-Lahham et al. showed that 3 mM of propionic acid significantly decreased LPS-stimulated TNF-α response in ATMs [33]. This study did not observe an inhibitory effect from propionate on TNF-α production but did observe a decrease in IL-6 release [33]. Evidence from our model and other experiments provides strong support for SFCA mediation in LPS-induced inflammation, suggesting the potential for SCFAs as a novel aid for the treatment of systemic inflammation, particularly in conditions where inflammation is exaggerated such as obesity.

FFAR receptors, FFAR2 and FFAR3, thought to be main receptors of SCFAs, are upregulated by LPS stimulation in monocytes and macrophages, suggesting a role in systemic inflammation, whereby upon activation SCFAs may trigger downstream anti-inflammatory signalling cascades [6,16]. This study revealed FFAR3 mRNA expression in acetate treated macrophages was significantly upregulated compared to LPS-stimulated macrophages. In LPS-stimulated monocytes and macrophages, acetate treatment trended towards a significant increase in FFAR2 mRNA expression compared to untreated, LPS stimulated cells, whilst a similar trend was observed in FFAR3 mRNA expression following acetate treatment in monocytes compared to LPS-stimulated-only monocytes.

The smaller sample size of acetate treatments within this model may have limited the capacity to achieve these statisically significant findings.

Evidence within the literature supports the ability of acetate to upregulate FFAR2 and FFAR3 receptors in human monocytes [16,34,35]. In contrast to acetate, our results showed that butyrate significantly decreased FFAR2 mRNA expression. Findings from this study and within the literature suggest that the SCFAs, particularly acetate, can affect FFARs, with downstream reductions in inflammatory pathways in obese subjects. Future studies investigating the downstream effects of SCFA-activated FFARs are required to fully elucidate the pathways in which FFARs inhibit systemic inflammation.

Another mechanism in which SCFAs have been suggested to decrease inflammation, is via inhibition of HDACs, whereby HDAC inhibition is proposed to affect downstream signalling of NF-κB and MAPK inflammatory pathways. Butyrate and propionate are believed to be the most potent HDAC inhibitors [17]. Data from this study support this, where propionate and butyrate significantly inhibited mRNA expression of HDAC9 in both monocytes and macrophages, whilst acetate and butyrate treatment decreased HDAC2 mRNA expression in monocytes. Similarly, Usami et al. confirmed that butyrate and propionate reduced LPS-induced TNF-α production in mononuclear cells through inhibition of NF-κB, in a similar fashion to trichostatin A, a HDAC inhibitor [36]. Interestingly, Aoyama et al. highlighted that these effects were seen in the absence of FFAR2 and FFAR3 activation, suggesting that HDAC inhibition pathways can operate independently to FFAR mediated pathways and that the NF-κB pathway may be the primary pathway affected by HDAC inhibition [37].

It is proposed that HDAC1 and HDAC2 association with p65 is linked to NF-κB expression, where HDAC inhibition reduces expression of NF-κB regulated genes [38]. Results from this study partly agree, as seen by a significant decrease in HDAC2 mRNA expression in monocytes following acetate and butyrate treatments. In contrast, we observed an increase in HDAC1 expression following acetate and butyrate treatment in monocytes. Ashburner et al. found that overexpression of either HDAC1 or HDAC2, reduced TNF-α induced expression of an NF-κB-dependent reporter gene (3XκB-Luc) [38]. Upregulation of HDAC1 expression in this study may support the role of HDAC1 by this mechanism, whilst results indicated that HDAC2 may have a similar role to HDAC9, whereby inhibition reduces expression of NF-κB regulated genes. Multiple linear regression analysis of our data revealed that mRNA expression of HDACs 1 and 9 predicted mRNA expression of NF-κB-related genes NFKB1 and RELA in LPS-stimulated monocytes following propionate (Appendix A) and butyrate (Appendix A) treatment. These models highlighted significant relationships between downregulation of HDAC1 with downregulation of NFKB1 following propionate treatment and downregulation of HDAC1 and HDAC2 with downregulation of RELA following butyrate treatment in LPS-stimulated monocytes. These models indicate that SCFA (specifically propionate and butyrate)-mediated HDAC inhibition plays a clear role in the attenuation of the NF-κB pathway.

HDAC inhibition has also been linked to the MAPK pathway, as inhibition of HDAC1-3 has been hypothesised to inhibit the MAPK pathway. Jeong et al. found that inhibition of HDAC1-3 decreased LPS-induced phosphorylation of p38MAPK and subsequently LPS-induced TNF-α and IL-1β expression [39]. Indeed, multiple linear regression models indicated that downregulation of MAPK1 mRNA expression in LPS-stimulated monocytes was predicted by inhibition of HDAC2 following propionate treatment (Appendix A) and inhibition of HDAC2 and HDAC9 post butyrate (Appendix A) SCFA treatment. These models suggest that propionate and butyrate mediated inhibition of HDACs plays a significant role in downregulation of the MAPK pro-inflammatory pathway.

Lastly, SCFAs may directly affect the inflammatory pathways downstream of MAPK and NF-κB via passive diffusion into the cell [6]. SCFAs may have a direct effect on the NF-κB p65/p50 heterodimer. This study highlighted that butyrate and acetate in monocytes and acetate, butyrate and propionate in macrophages, significantly reduced expression of NFKB1-related gene p50. Additionally, butyrate in monocytes and acetate in macrophages, decreased expression of RELA associated gene p65. Furthermore, results indicated that butyrate and propionate had the ability to decrease expression of MAPK1 associated genes. Whether these effects are direct or indirect, as a result of FFAR activation or HDAC inhibition, remains unclear.

A few limitations exist within this study. Small sample size within the healthy control group may have limited potential observations. It is important to note, however, that few studies have examined the differences in pro-inflammatory gene expression between healthy, obese and post-weight loss subjects. Therefore, the observations in this study were novel and provide groundwork for future studies to build on. Furthermore, a limitation existed in the isolation of macrophages from adipose tissue. The vascularisation and variation in visceral adipose tissue played a limiting factor in the number of macrophages that could be isolated per sample. This limited the number of SCFA treatments that could be performed on each sample and may have limited the number of significant observations. Another challenge in working with ATMs is that they were not as responsive to LPS stimulation as monocytes. Therefore, effects of SCFA treatments in ATMs were more subtle compared to the effects seen in monocytes. Whilst there were limitations to working with ATMs, our experiments are important, as adipose tissue is difficult to access and provides vital information that reflects the host environment in obesity. To date limited research has examined the effects of SCFAs on inflammation in human ATMs, making this work particularly novel.

In summary, this study provided evidence for two strategies to attenuate chronic low-grade systemic inflammation in obesity. Findings demonstrated that characteristics of systemic inflammation in obesity, including elevated pro-inflammatory cytokine production (TNF-α and IL-6) and upregulated NF-κB-related genes (NFKB1 and RELA), present in obesity were reduced by significant weight loss and ex vivo SCFA treatment. Weight loss was successful in reducing these biomarkers of systemic inflammation, confirmed by reductions in circulating pro-inflammatory cytokine TNF-α and downregulation in NF-κB-related genes, NFKB1 and RELA. Further, findings highlighted the anti-inflammatory effects of SCFAs in attenuating systemic inflammation in obesity, characterised by reductions in pro-inflammatory cytokines TNF-α and IL-6 and downregulation of pro-inflammatory NF-κB- and MAPK-related genes, NFKB1, RELA and MAPK1 in obese immune cells. Furthermore, results elucidated a number of mechanisms of SCFA action, such as inhibition of HDACs and activation of FFARs. In conclusion, this study highlights two effective strategies for attenuating systemic inflammation in obesity, that could be utilised to reduce the burden of obesity-associated chronic disease.

## Figures and Tables

**Figure 1 nutrients-14-00765-f001:**
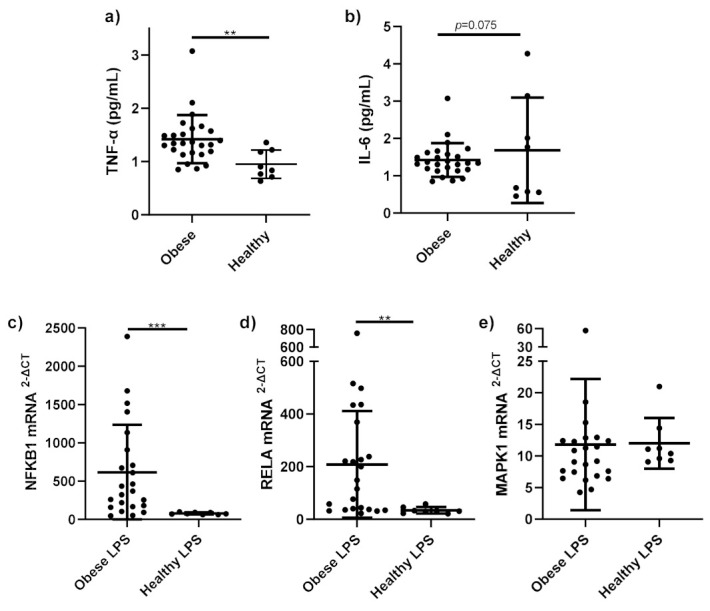
Pro-inflammatory cytokines and gene expression in obese and healthy weight subjects. Serum pro-inflammatory cytokines (**a**) TNF-α (pg/mL); (**b**) IL-6 (pg/mL) and mRNA expression of (**c**) NFκB1; (**d**) RELA; (**e**) MAPK1 in monocytes stimulated with LPS (1 ng/mL) and incubated for 15 h, were measured in obese (*n* = 25) and healthy weight subjects (*n* = 8). Parametric data analysed by RM one-way ANOVA and post hoc analysis Tukey’s multiple comparison test, non-parametric by the Friedman test and post hoc analysis by Dunn’s multiple comparisons test. TNF-α; Tumour necrosis factor-alpha, IL-6; interleukin 6. ** *p* < 0.01, *** *p* < 0.001. Data presented as the mean (SD).

**Figure 2 nutrients-14-00765-f002:**
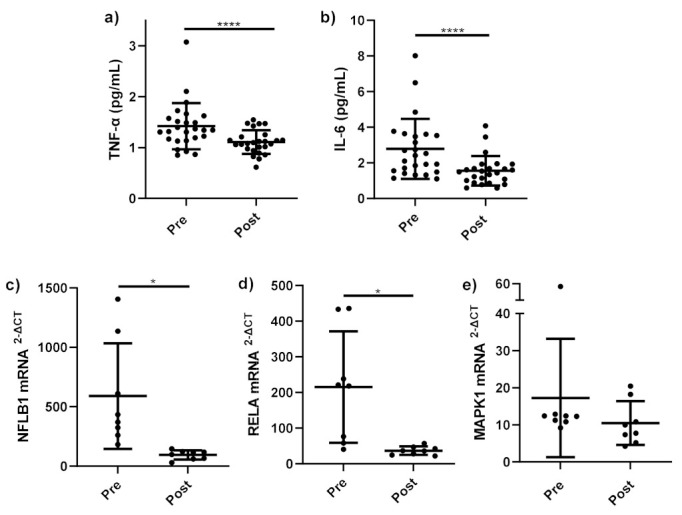
Pro-inflammatory cytokines and gene expression before and after bariatric surgery. Serum pro-inflammatory cytokines (**a**) TNF-α (pg/mL) (*n* = 25); (**b**) IL-6 (pg/mL) (*n* = 25) and mRNA expression of (**c**) NFκB1 (*n* = 8), (**d**) RELA (*n* = 8), and (**e**) MAPK1 (*n* = 8) in monocytes stimulated with LPS (1 ng/mL) and incubated for 15 h, measured before bariatric surgery (Pre) and 12–18 months follow up (Post). Parametric data analysed by RM one-way ANOVA and post hoc analysis Tukey’s multiple comparison test, non-parametric by the Friedman test and post hoc analysis by Dunn’s multiple comparison test. TNF-α: tumour necrosis factor-alpha, IL-6; interleukin 6, Pre (pre-operative), Post (12–18 months follow up), LPS; lipopolysaccharide. * *p* < 0.05, **** *p* < 0.0001. Data presented as the mean (SD).

**Figure 3 nutrients-14-00765-f003:**
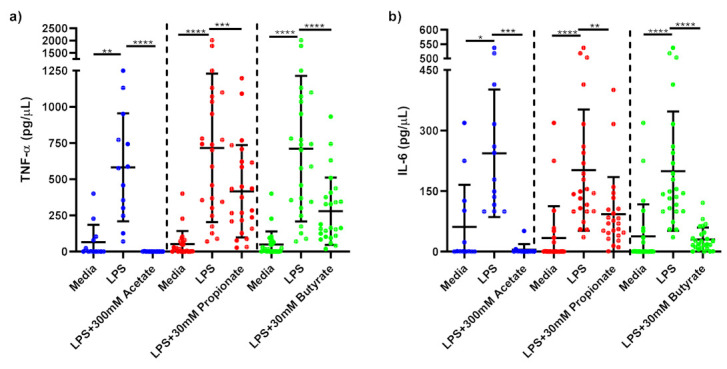
Cytokine response of LPS-stimulated monocytes following short-chain fatty acid treatments. Cell culture supernatant concentrations of (**a**) TNF-α (pg/µL); (**b**) IL-6 produced by monocytes stimulated with LPS (1 ng/mL) alone or treated simultaneously with sodium acetate (*n* = 13) 300 mM or sodium propionate (*n* = 24) 30 mM or sodium butyrate (*n* = 25) 30 mM following 15 h incubation. Parametric data analysed by RM one-way ANOVA and post hoc analysis Tukey’s multiple comparison test, non-parametric by the Friedman test and post hoc analysis by Dunn’s multiple comparisons test. TNF-α; tumour necrosis factor-α, IL-6: interleukin 6, LPS; lipopolysaccharide, Ac; acetate, Pr; propionate, Bu; butyrate. * *p* < 0.05, ** *p* < 0.01, *** *p* < 0.001 and **** *p* < 0.0001. Data expressed as the mean (SD).

**Figure 4 nutrients-14-00765-f004:**
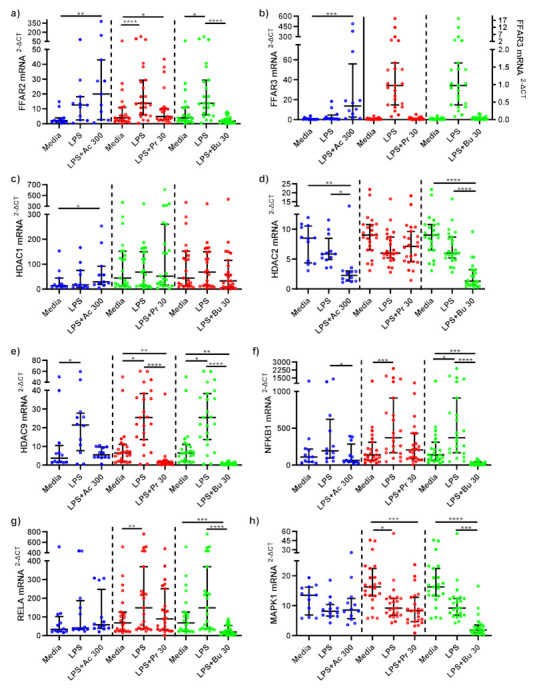
mRNA expression response of LPS-stimulated monocytes following short-chain fatty acid treatments. mRNA expression of (**a**) FFAR2, (**b**) FFAR3 (sodium acetate [left Y axis], sodium propionate [right Y axis] and sodium butyrate [right Y axis]), (**c**) HDAC1, (**d**) HDAC2, (**e**) HDAC9, (**f**) NFKB1, (**g**) RELA, and (**h**) MAPK1 produced by monocytes stimulated with LPS (1 ng/mL) alone or treated simultaneously with sodium acetate (*n* = 13) 300 mM or sodium propionate (*n* = 24) 30 mM or sodium butyrate (*n* = 25) 30 mM following 15 h incubation. Parametric data analysed by RM one-way ANOVA and post hoc analysis Tukey’s multiple comparison test, non-parametric by the Friedman test and post hoc analysis by Dunn’s multiple comparisons test. LPS; lipopolysaccharide, Ac; acetate, Pr; propionate, Bu; butyrate. * *p* < 0.05, ** *p* < 0.01, *** *p* < 0.001 and **** *p* < 0.0001. Data expressed as the mean (SD).

**Figure 5 nutrients-14-00765-f005:**
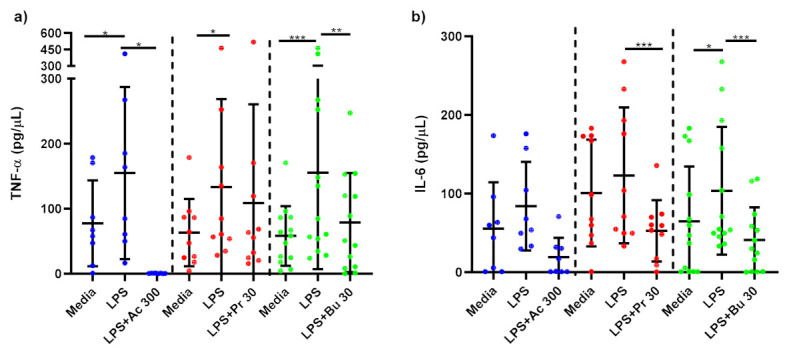
Cytokine response of LPS-stimulated adipose tissue macrophages following short-chain fatty acid treatments. Cell culture supernatant concentrations of (**a**) TNF-α (pg/µL); (**b**) IL-6 produced by macrophages stimulated with LPS (1 ng/mL) alone or treated simultaneously with sodium acetate (*n* = 10) 300 mM or sodium propionate (*n* = 11) 30 mM or sodium butyrate (*n* = 13) 30 mM following 15 h incubation. Parametric data analysed by RM one-way ANOVA and post hoc analysis Tukey’s multiple comparison test, non-parametric by the Friedman test and post hoc analysis by Dunn’s multiple comparisons test. TNF-α; tumour necrosis factor-α, IL-6: interleukin 6, LPS; lipopolysaccharide, Ac; acetate, Pr; propionate, Bu; butyrate. * *p* < 0.05, ** *p* < 0.01 and *** *p* < 0.001. Data expressed as the mean (SD).

**Figure 6 nutrients-14-00765-f006:**
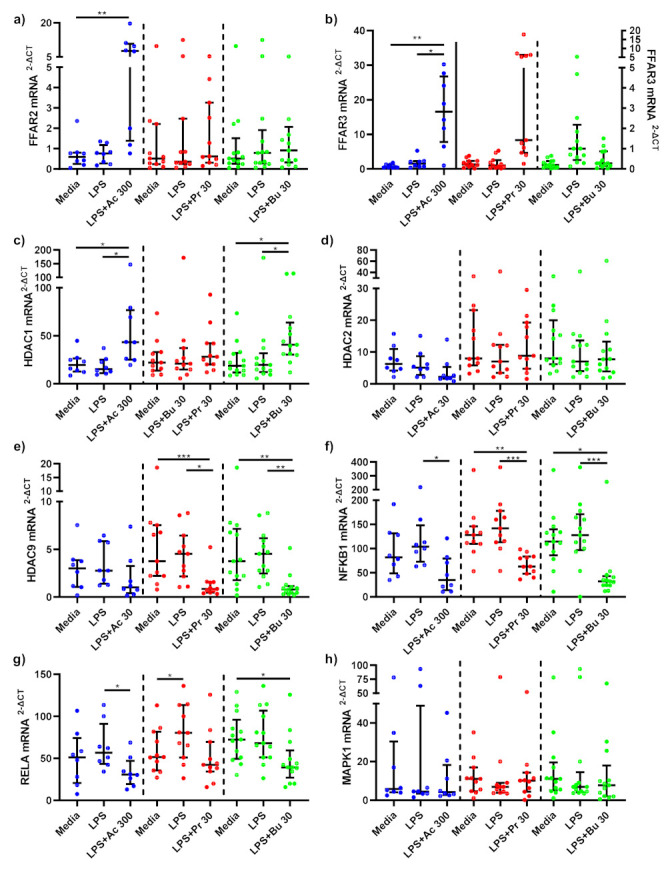
mRNA expression response of LPS-stimulated adipose tissue macrophages following short-chain fatty acid treatments. mRNA expression of (**a**) FFAR2, (**b**) FFAR3, (**c**) HDAC1, (**d**) HDAC2, (**e**) HDAC9, (**f**) NFKB1, (**g**) RELA, and (**h**) MAPK1 produced by macrophages stimulated with LPS (1 ng/mL) alone or treated simultaneously with sodium acetate (*n* = 10) 300 mM or sodium propionate (*n* = 11) 30 mM or sodium butyrate (*n* = 13) 30 mM following 15 h incubation. Parametric data analysed by RM one-way ANOVA and post hoc analysis Tukey’s multiple comparison test, non-parametric by the Friedman test and post hoc analysis by Dunn’s multiple comparisons test. LPS; lipopolysaccharide, Ac; acetate, Pr; propionate, Bu; butyrate. * *p* < 0.05, ** *p* < 0.01 and *** *p* < 0.001. Data expressed as the mean (SD).

**Table 1 nutrients-14-00765-t001:** Subject Characteristics.

	Obese (*n* = 43)	Healthy (*n* = 8)	*p*-Value ^1^	Follow-Up (*n* = 26)	*p*-Value ^1^
Demographics					
Age (years)	41.6 ± 9.5	28.9 ± 5.9	<0.001	42.77 ± 8.99	
Gender (Female %)	36 (84%)	6 (75%)	0.562	22 (85%)	
Height (cm)	165.6 ± 8.6	171.3 ± 7.7	0.086	1.65 ± 0.08	
Average Follow-Up Time (months)				15.69 ± 1.78	
Body composition					
Weight (kg)	127.4 [107.3, 151.6]	64.3 [62.5, 71.35]	<0.001	95.58 ± 23.28	<0.001
BMI (kg/m^2^)	46.4 [38.3, 54.0]	22.3 [21.7, 24.1]	<0.001	35.10 ± 7.31	<0.001
Android fat (%)	60.19 [57.27, 62.25] *	27.59 [18.47, 35.89]	<0.001	41.54 ± 8.50	<0.001
Gynoid Fat (%)	54.52 [49.02, 57.72] *	27.83 [24.20, 37.07]	<0.001	46.91 ± 8.84	<0.001
Total Body Fat (%)	52.6 [48.3, 55.7] *	26.8 [25.2, 31.3]	<0.001	42.61 ± 9.46	<0.001
Total Body Lean (%)	44.8 [42.7, 42.7] *	69.4 [64.9, 72.1]	<0.001	56.25 ± 8.00	<0.001

BMI; Body Mass Index, * *n* = 32. Data presented as the mean ± SD or the median (interquartile range). ^1^ vs. obese.

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
