# Peer review of "Weight Loss and Short-Chain Fatty Acids Reduce Systemic Inflammation in Monocytes and Adipose Tissue Macrophages from Obese Subjects"

_nutrients, 2022, doi:10.3390/nu14040765_

Round 1

Reviewer 1 Report

In the present manuscript, the authors sought to evaluate the effect of weight loss on the proinflammatory profile in obese subjects, as well as the potential use of ex-vivo short-chain fatty acids as a pro-inflammatory treatment for systemic inflammation in obese subjects.

The authors did a fantastic job providing background information and setting up the need for the study.

-For some reason, figure is cut and can´t be properly appreciated. This must be corrected.

-In my opinion, figures 3 and 4 are too loaded. I´d recommend splitting each into two figures to make the information more digestible.

-I´d recommend including a discussion the findings in the following reference

            Consumption of a high energy density diet triggers microbiota dysbiosis, hepatic lipidosis, and microglia activation in the nucleus of the solitary tract in rats

Author Response

For some reason, figure is cut and can´t be properly appreciated. This must be corrected.

Thank you, Figure 1 has been revised to include all graphs appropriately.

-In my opinion, figures 3 and 4 are too loaded. I´d recommend splitting each into two figures to make the information more digestible.

Response: As per reviewer’s suggestion, figures 3 and 4 have been split. Figures have been split in to cytokine response (Figures 3 & 5) and mRNA response (Figures 4 & 6). Furthermore, all graphs have been edited to fit within journal margins.

-I´d recommend including a discussion the findings in the following reference

            Consumption of a high energy density diet triggers microbiota dysbiosis, hepatic lipidosis, and microglia activation in the nucleus of the solitary tract in rats

Response: Thank you for this suggestion. This is an interesting study that highlights the effect of high energy diet in causing microbiota dysbiosis and consequently increasing circulating inflammatory markers.

Lines 463-465: Recent evidence from a study in mice revealed that gut dysbiosis, as a result of high energy diets, leads to an increase in circulating inflammatory markers.[27]

Reviewer 2 Report

REW: Weight loss and SCFA reduces systemic inflammation in monocytes and AT macrophages in obese subjects

Eslick et al. investigated the effect of obesity and weight loss achieved by bariatric surgery on the inflammatory status of circulating monocytes and isolated adipose tissue (AT) macrophages. The authors also looked at the effect of the short-chain fatty acids (SCFA) - acetate, propionate, and butyrate  - on LPS-stimulated inflammatory pathways in these monocytes/macrophages.

The study is novel and interesting in terms of the effect of a metabolic product of the gut microbiota on the inflammatory signaling of immune cells in blood and adipose tissue.

The clinical protocol is well designed, and the combination of comparison with healthy controls and longitudinal investigation of obese subject after bariatric surgery represents a suitable approach to investigate this issue. However, I found several major problems that should be adressed.

Introduction

- some of the references in the introduction do not seem appropriate, e.g.: ref 8 does not show a decrease in systemic inflammation after bariatric surgery; ref 12 does not show that SCFAs are produced by fermenting microbiota; ref 13 does not show that SCFAs are produced in the proximal segments of the colon. I suggest the authors carefully check all references in the article.

Methods

- In the paragraph "design of the study", the number of subjects should be indicated

- the number of lean controls is really low when compared to obese. Why the authors did not include more lean subjects?

Results

- It is not clear to me, why 3 different experiments were perfomed with SCFA? I would expect setting of experiment with control (media only), LSP stimulation, LPS + acetate, LPS + butyrate, LPS + propionate. This would be more appropriate in my point of view.

- Presentation and interpretation of results seems to me very problematic :

-  in some cases the statistical significance and the comment does not seem to match the data- e.g.: line 59-61.. does not correspond to the picture

- Line 63-64 ..propionate nad butyrate did not significanrly change FFAR3 mRNA expression … how is this possible, when the clear decrease is present in LPS+propionate, and LPS-butyrare when compareed to LPS-induction (Fig 3d). Similar effect shows Fig 3g – where the strong significances are present. The same applies to the line 72-77.

-  Line 108-109, 117-118 .. the statements in the results does not correspond to the figure 4!!

I suggest to check very carefully the whole statistics of these experiments, the presentation of the figures and the commentary of the results !! I also suggest to shorten the commetary of results to make it more clear and understandable.

Discussion

- is well writen and understandable, but again there are discrepancies between the claims and the results shown in the figures (e.g. line 228-229, 275-276)

Minor points:

- Fig. 1e) is not inserted correctly

- Fig. 3 a), b) – there are no descriptions of y-axes

- Fig. 4 – the heading should be corrected – I expect that the data are from AT macrophages, but no monocytes

- Please specify that monocytes are from blood and macrophages from AT in paragraph headings in the results section

- All figures – please specify the statistical method used for the data and p-value generation

- Abberviation „ATM“ in the discussion should be specified at the first mention in the text.

Round 2

Reviewer 2 Report

I appreciate the study design, methods and results. The authors performed very nice study. The authors responded also sufficiently to almost all the comments and improved significantly the manuscript. However, some points are still necessary to be improved.

1) From the first round of review:

REW: It is not clear to me, why 3 different experiments were perfomed with SCFA? I would expect setting of experiment with control (media only), LSP stimulation, LPS + acetate, LPS + butyrate, LPS + propionate. This would be more appropriate in my point of view.

Authors Response: Indeed the setting of the experiment is control (media only), LPS stimulation, LPS + acetate, LPS + butyrate, LPS + propionate. Media only and LPS only are listed for each SCFA type as a different number of samples were treated in pairs i.e. In macrophages, samples sizes for each SCFA type were sodium acetate (n=10) 300mM, Sodium propionate (n=11) and Sodium Butyrate (n=13) 30mM. The reason for this is that in some instances an inadequate number of cells were isolated from samples, therefore all SCFA conditions could not be performed in the same experiment.

REW 2nd round: I probably still don’t understand the setting of the protocol with SCFA. You applied pairwise comparisons – so that 1) you have the same cells for control , LPS and LPS+ acetate and the other cells for control , LPS and LPS+ butyrate? Or you have 2) the same cells for control, LPS, LPS + acetate, LPS + butyrate and LPS+ propionate, while for some experiments some data/cells in the group is missing (e.g. LPS-acetate)?  If option 1 is valid, then everything is probably fine, but if the experiment was conducted according to option 2, the statistics should be processed using one-way ANOVA for all 5 groups at once and the results should be presented in one graph. I cannot imagine how you would select controls and LPS-treated cells for comparison with LPS-acetate in this type of experiment.  How would you know which control data not to use to apply pairwise comparison?

2)  I found again a mistake in comments of the results:

line 363-364:  ... "Acetate significantly downregulated HDAC1 mRNA expression compared to both LPS-stimulated only (p=0.040) and media-only (p=0.038) macrophages (Figure 6c)."  - however,  according to Fig 6c acetate UP-REGULATED mRNA of HDAC1 

3) The part of the results with SCFA treatment is still very complicated and becomes almost incomprehensible for the reader.  Even the authors themselves have several times had errors in the commentary of the results, probably due to the complexity of the commentary. I strongly recommend rewriting this section as simply as possible. It is not necessary to repeat all the significances and results that are found in the figures, but rather to highlight the unambiguous message.

Author Response

Nutrients – Reviewer 2 comments

REW 2nd round: I probably still don’t understand the setting of the protocol with SCFA. You applied pairwise comparisons – so that 1) you have the same cells for control , LPS and LPS+ acetate and the other cells for control , LPS and LPS+ butyrate? Or you have 2) the same cells for control, LPS, LPS + acetate, LPS + butyrate and LPS+ propionate, while for some experiments some data/cells in the group is missing (e.g. LPS-acetate)?  If option 1 is valid, then everything is probably fine, but if the experiment was conducted according to option 2, the statistics should be processed using one-way ANOVA for all 5 groups at once and the results should be presented in one graph. I cannot imagine how you would select controls and LPS-treated cells for comparison with LPS-acetate in this type of experiment.  How would you know which control data not to use to apply pairwise comparison?

Response: The setting of the experiment is as described in option 1. Pairwise comparisons were made with cells from the same individual for control, LPS and LPS+acetate and other cells for control, LPS and LPS+butyrate.

2)  I found again a mistake in comments of the results:

line 363-364:  ... "Acetate significantly downregulated HDAC1 mRNA expression compared to both LPS-stimulated only (p=0.040) and media-only (p=0.038) macrophages (Figure 6c)."  - however,  according to Fig 6c acetate UP-REGULATED mRNA of HDAC1 

Response: Indeed, this result is supposed to have said upregulated. This has been amended.

3) The part of the results with SCFA treatment is still very complicated and becomes almost incomprehensible for the reader.  Even the authors themselves have several times had errors in the commentary of the results, probably due to the complexity of the commentary. I strongly recommend rewriting this section as simply as possible. It is not necessary to repeat all the significances and results that are found in the figures, but rather to highlight the unambiguous message.

Response: The results section has been rewritten as per request. Results section was rewritten to discuss significant results pertaining to LPS-stimulated only and LPS + SCFA treated cells as to minimise commentary and to highlight a more consistent message.

Lines: 332-357: LPS-stimulated monocytes treated with acetate and propionate did not signifi-cantly change FFAR2 mRNA gene expression compared to untreated, LPS-stimulated monocytes (Figure 4a). FFAR2 mRNA expression was however, significantly decreased in LPS-stimulated monocytes treated with butyrate compared to LPS-stimulated only monocytes (p<0.001). A non-significant increase in FFAR3 mRNA expression was ob-served in LPS-stimulated monocytes treated with acetate (p=0.124) compared to LPS-stimulated only monocytes, whilst propionate and butyrate did not significantly change FFAR3 mRNA expression (Figure 4b).

SCFAs did not alter HDAC1 mRNA expression compared to LPS-stimulated only monocytes (Figure 4c). Acetate (p=0.013) and butyrate (p<0.001) both significantly re-duced HDAC2 mRNA expression compared to LPS-stimulated only monocytes, whilst propionate had no effect (Figure 4d). LPS stimulation significantly increased HDAC9 mRNA expression in monocytes (p<0.05)(Figure 4e). Acetate treatment did not affect mRNA expression of HDAC9, whilst propionate (p<0.001) and butyrate (p<0.001) treatments significantly decreased LPS-induced HDAC9 expression compared to LPS-stimulated only monocytes.

NF-κB associated gene, NFKB1 mRNA expression was downregulated in LPS-stimulated monocytes treated with acetate (p=0.013) and butyrate (p<0.001) compared to LPS-stimulated only monocytes (Figure 4f). LPS-stimulated monocytes treated with propionate did not change NFKB1 mRNA gene expression. NF-κB associ-ated gene RELA mRNA expression remained unchanged following acetate and pro-pionate SCFA treatments (Figure 4g). RELA mRNA gene expression significantly de-creased in LPS-stimulated monocytes treated with butyrate compared to LPS-stimulated only monocytes (p<0.001).

mRNA gene expression of MAPK1 was not affected by acetate or propionate treatments compared to LPS-stimulated only monocytes (Figure 4h). Butyrate treat-ment significantly decreased MAPK1 compared to LPS-stimulated only (p<0.001) monocytes.

Lines: 389-411: In LPS-stimulated macrophages, acetate treatment trended towards a significant increase in FFAR2 mRNA expression compared to LPS-stimulated only monocytes (p=0.0733)(Figure 6a). No changes in FFAR2 mRNA expression was seen as a result of propionate and butyrate treatments. Acetate treatment significantly upregulated FFAR3 mRNA expression compared to LPS-stimulated only macrophages (p=0.010) (Figure 6b). Similar to FFAR2, FFAR3 mRNA expression was not changed following propionate and butyrate SCFA treatments.  

 Acetate significantly upregulated HDAC1 mRNA expression compared to LPS-stimulated only (p=0.040) macrophages (Figure 6c). HDAC1 mRNA expression was not altered by propionate treatment. In contrast, HDAC1 mRNA expression was significantly higher in LPS-stimulated macrophages treated with butyrate compared to LPS-stimulated only (p=0.018) macrophages. HDAC2 mRNA expression was un-changed as a result of SCFA treatments (Figure 6d). HDAC9 mRNA expression was not affected by acetate, whilst propionate (p=0.017) and butyrate (p=0.007) significantly downregulated HDAC9 mRNA expression compared to LPS-stimulated only macro-phages (Figure 6e).

Acetate (p=0.043), propionate (p=0.002) and butyrate (p=0.006) SCFA treatments all significantly downregulated NF-κB associated gene NFKB1 in LPS-stimulated macrophages compared to LPS-stimulated only macrophages (Figure 6f). NF-κB asso-ciated gene RELA mRNA expression in LPS-stimulated macrophages was significantly reduced by acetate treatment compared to LPS-stimulated only macrophages (p=0.018), whilst propionate and butyrate had no effect (Figure 6g). Finally, MAPK associated gene MAPK1 mRNA gene expression in LPS-stimulated macrophages was not altered by SCFA treatment (Figure 6h).
